# Diagnosing and Remedying Knowledge Deficiencies in LLMs via Label-free Curricular Meaningful Learning

**Kai Xiong**[1,2]   **Xiao Ding**[1*]   **Yixin Cao**[3]   **Li Du**[4]   **Jiahao Ying**[5]   **Yang Zhao**[1]
**Bing Qin**[1]   **Ting Liu**[1]

[1]Research Center for Social Computing and Interactive Robotics
  Harbin Institute of Technology, Harbin, China
[2]Zhongguancun Laboratory, Beijing, China
[3]Institute of Trustworthy Embodied AI, Fudan University, Shanghai, China
[4]Beijing Academy of Artificial Intelligence, Beijing, China
[5] Singapore Management University, Singapore
`{kxiong, xding, yangzhao, qinb, tliu}@ir.hit.edu.cn`
`yxcao@fudan.edu.cn`

## Abstract

Large Language Models (LLMs) have demonstrated impressive generalization ability by learning from extensive unlabeled text. However, they still exhibit reasoning mistakes, which can affect their trustworthiness and reliability. Although users can interact with LLMs and provide diverse and comprehensive queries to expose the flaws of LLMs, obtaining sufficient and effective feedback is demanding. Furthermore, comprehensively evaluating LLMs with limited labeled samples is difficult. These make it a challenge to diagnose and remedy the deficiencies in LLMs through rich label-free user queries. To tackle this challenge and considersing that LLMs' reasoning mistakes often stem from knowledge deficiencies, we propose label-free curricular meaningful learning (LaMer), which first employs relative entropy to diagnose and quantify knowledge deficiencies of LLMs in a label-free setting. Then, LaMer adaptively synthesizes augmentation data based on deficiency severity and progressively remedies them with a curricular remedy strategy. Experiments show that LaMer effectively diagnoses and remedies knowledge deficiencies in LLMs, improving various LLMs across seven out-of-distribution (OOD) reasoning benchmarks, achieving comparable results to baselines with only 40% training data. LaMer even surpasses methods that rely on labeled data for deficiency diagnosis. In application, LaMer offers a diagnostic tool for efficient LLM development. The code is available at `https://github.com/Waste-Wood/LaMer`.

## 1 Introduction

Large language models (LLMs) have made significant advancements in various fields recently (Kim et al., 2024; Li et al., 2024a). By implicitly mining and learning information from vast amounts of unlabeled text via language modeling, LLMs have demonstrated remarkable generalization abilities. This enables them to answer various user queries across many applications such as reasoning (Tang et al., 2024; Xiong et al., 2024) and recommender systems (Lei et al., 2023; Wu et al., 2024). However, despite their potential, LLMs still have limitations. Due to their statistical nature, LLMs occasionally make reasoning mistakes (Jung et al., 2022; Wang et al., 2023), which can undermine user trust and the reliability of their applications. A significant challenge is that the knowledge mining process is implicit, making it difficult to discern what LLMs are particularly good or bad at. This lack of transparency hinders targeted improvements and quality assurance of LLMs. Additionally, relying on users for sufficient and effective feedback is often difficult and impractical, as it requires extra effort and users typically seek answers to questions they do not fully understand. This situation poses a significant obstacle in continually improving LLMs based on massive label-free user queries.

---

*Corresponding Author

To enhance LLMs, current researches predominantly follow two methodologies: unsupervised language modeling (Fujii et al., 2024; Guo et al., 2024b) and supervised fine-tuning (SFT) (Mitra et al., 2023; Xu et al., 2024), which are respectively shown in Figure 1 (a) and (b). Unsupervised language modeling uses vast amounts of unlabeled data, enabling LLMs to learn knowledge implicitly. In contrast, SFT involves training LLMs on labeled datasets tailored to specific tasks. Despite their advantages, both approaches have limitations. First, they can be inefficient as it necessitates the inclusion of extensive data indiscriminately, which may not address enough reasoning mistakes of LLMs. Second, they still lack a comprehensive understanding of LLMs, leading to the inability to make targeted improvements. As a result, this further leads to ineffectiveness in addressing specific and long-tail questions. Moreover, labeling user queries can help to reveal some mistakes of LLMs, while it is costly and challenging to use limited labeled samples to thoroughly evaluate LLMs that generalize well (Liu et al., 2023; Zheng et al., 2024). These limitations highlight the need for more efficient and cost-effective methods to diagnose and improve LLMs.

To tackle the above challenges, and considering that reasoning mistakes in LLMs often arise from knowledge deficiencies such as lack of knowledge and ineffective application of existing knowledge (Gao et al., 2023; Xiong et al., 2024), we aim to diagnose the knowledge deficiencies in LLMs and remedy them without any costly annotations. By diagnosing these knowledge deficiencies, we can tailor solutions for targeted and efficient improvements. Furthermore, our method enables LLMs to evolve continuously with increasing user engagement, even in the absence of user feedback. This approach ensures that LLMs remain current and adaptable, capable of handling specific and long-tail user demands, avoiding costly data labeling processes, and promoting a more efficient online development cycle. In this paper, as shown in Figure 1 (c), we design an indirect supervision method called label-free curricular meaningful learning (LaMer), which first diagnoses the knowledge deficiencies of a specific LLM in a label-free setting, and then devise curricular meaningful learning to efficiently and effectively remedy the deficiencies of corresponding LLM. Specifically, LaMer first extracts relevant knowledge for user queries. Subsequently, inspired by information theory (Shannon, 1948), where relative entropy (Kullback & Leibler, 1951) can estimate the extra information needed to change from a distribution to another. We leverage relative entropy with the extracted knowledge to diagnose the knowledge deficiencies of LLMs without relying on labels. Finally, we adopt curricular meaningful learning, which initially uses meaningful learning (Xiong et al., 2024) to adaptively synthesize augmented data across various scenarios based on the severity of the deficiencies, and then employ curricular deficiency remedy to progressively address these deficiencies from minor to severe.

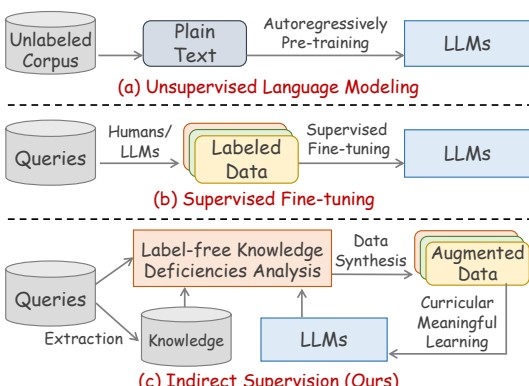

Figure 1: (a) Unsupervised language modeling, (b) supervised fine-tuning, and (c) our proposed indirect supervision method.

We conduct extensive experiments on 4 open-source LLMs and evaluate LaMer and baselines across 7 out-of-distribution (OOD) reasoning benchmarks. The results show that LaMer proficiently diagnoses the knowledge deficiencies in various LLMs, leading to more efficient and effective improvements compared to baselines. It achieves comparable results to baselines with only 40% training data. Further analyses reveal that LaMer not only surprisingly surpasses methods relying on labeled data to detect deficiencies but also highlights its efficiency and effectiveness in diagnosing and remedying knowledge deficiencies in LLMs, offering a more robust solution for improving their application. We summarize our contributions as follows:

- We incorporate relative entropy to effectively diagnose knowledge deficiencies in LLMs without labels, breaking the limitation of relying solely on existing labeled datasets.

- We develop curricular meaningful learning to remedy the knowledge deficiencies in LLMs.

- Our proposed LaMer excels in diagnosing and remedying the knowledge deficiencies, towards maximizing the potential of existing open-source LLMs.

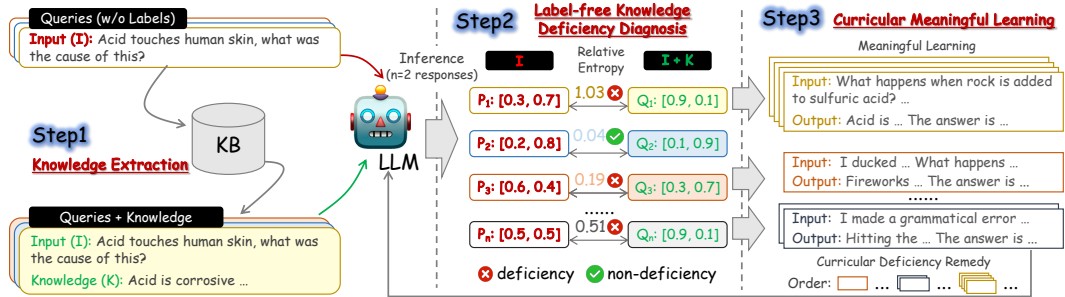

Figure 2: The whole workflow of LaMer to automatically diagnose and remedy the knowledge deficiencies of LLMs in a label-free setting.

## 2 METHOD: LABEL-FREE CURRICULAR MEANINGFUL LEARNING

We design a label-free curricular meaningful learning framework named LaMer, which utilizes user queries to efficiently diagnose and remedy the knowledge deficiencies in LLMs without labels. As illustrated in Figure 2, LaMer consists of 3 steps: (1) **Knowledge Extraction** obtains relevant knowledge from an external knowledge base for each query to help diagnose knowledge deficiencies; (2) **Label-free Knowledge Deficiency Diagnosis** leverages relative entropy to automatically diagnose and quantify the deficiencies in LLMs, which does not rely on labels; (3) **Curricular Meaningful Learning** incorporates the idea of human conducting meaningful learning (Tenenbaum, 2018) to first adaptively synthesize examples in various scenarios for each knowledge deficiency, and then utilize a curricular remedy strategy to effectively repair the knowledge deficiencies from minor to severe.

### 2.1 KNOWLEDGE EXTRACTION

To acquire knowledge for the following deficiency diagnosis step, we employ an external knowledge base GenericsKB (Bhakthavatsalam et al., 2020) to obtain knowledge for each query $d$ in the given query set $D$. GenericsKB is a large-scale resource with more than 3.4M naturally occurring generic facts (such as "Trees remove carbon dioxide from the atmosphere" and "Dog barks").

Specifically, to ensure the quality of GenericsKB, we first filter out facts with a confidence score lower than 0.7 and remove duplicates. Next, we employ FlagEmbedding (Zhang et al., 2023a) to represent the facts in GenericsKB and each query $d$ in $D$ as a dense embedding. Finally, for each query $d \in D$, we apply cosine similarity to match $m$ pieces of knowledge $K = \{k_1, \cdots, k_m\}$ from GenericsKB. The matched knowledge $K$ and example $d$ are used in step 2 to diagnose knowledge deficiencies in a specific LLM $\mathcal{L}$ through relative entropy.

### 2.2 LABEL-FREE KNOWLEDGE DEFICIENCY DIAGNOSIS

To diagnose the knowledge deficiencies of a specific LLM $\mathcal{L}$ in a label-free setting, we propose to use relative entropy (Kullback & Leibler, 1951). This measure quantifies the additional information needed to transition from one distribution to another. Thus, by computing the relative entropy (RE) between predictive distributions of $\mathcal{L}$ before and after the introduction of knowledge, we can estimate the volume of information that this knowledge imparts to $\mathcal{L}$. If $\mathcal{L}$ exhibits a high RE on this knowledge, it suggests that the model either lacks this knowledge or is unable to integrate it into its problem-solving processes, a knowledge deficiency in $\mathcal{L}$ is diagnosed.

Specifically, given a query $d \in D$, which possesses an input $I$. We first send $d$ to LLM $\mathcal{L}$ to obtain $n$ responses $O = \{o_1, o_2, \cdots, o_n\}$ and the negative log-likelihood (NLL) of each response $o_i$ conditioned on $I$. Hereafter, we acquire a prior distribution of $\mathcal{L}$ over $O$:

$$p_i = \mathcal{L}(o_i|I),$$
$$P = \text{Softmax}([p_1, \cdots, p_n]), \tag{1}$$

where $p_i$ denotes the NLL of response $o_i$ conditioned on $I$ according to $\mathcal{L}$. $P \in \mathbb{R}^n$ denotes the prior distribution of $\mathcal{L}$ over $O$. Softmax is the normalization function.

Secondly, for each extracted knowledge $k \in K$ of $d$, we additionally introduce $k$ to $\mathcal{L}$ to fetch a knowledge-based posterior distribution of $\mathcal{L}$ over $O$:

$$
\begin{aligned}
q_i &= \mathcal{L}(o_i | k, I), \\
Q &= \text{Softmax}([q_1, \cdots, q_n]),
\end{aligned}
\tag{2}
$$

where $q_i$ denotes the NLL of response $o_i$ after introducing $k$ to $\mathcal{L}$. $Q \in \mathbb{R}^n$ denotes the knowledge-based posterior distribution of $\mathcal{L}$ over $O$.

Next, we compute relative entropy RE between $P$ and $Q$ to quantify the difficiency of $\mathcal{L}$ on $k \in K$:

$$
\text{RE} = -\sum_{i=0}^{m} P_i \times (log(Q_i) - log(P_i)).
\tag{3}
$$

Finally, after estimating RE on each knowledge of each query based on $\mathcal{L}$, we filter the knowledge and corresponding queries that result in an RE larger than a threshold $\tau$. Each filtered knowledge and its associated query are treated as a unit, representing a knowledge deficiency in $\mathcal{L}$. RE is also a quantification of the knowledge deficiency in $\mathcal{L}$.

Note that there might be two situations: (1) **Helpful**: the extracted knowledge has a positive impact on $\mathcal{L}$, resulting in higher confidence in the correct answer; (2) **Misleading**: the obtained knowledge has a negative impact on $\mathcal{L}$, resulting in higher confidence in the wrong answer. We suppose that both situations can expose the knowledge deficiencies in $\mathcal{L}$. The first situation suggests $\mathcal{L}$ might not grasp this knowledge or cannot properly apply this knowledge to problem-solving, while the second situation indicates that $\mathcal{L}$ does understand this knowledge but is easily misled by it.

## 2.3 CURRICULAR MEANINGFUL LEARNING

Humans adopt meaningful learning (Tenenbaum, 2018; Xiong et al., 2024) to induce and learn new knowledge through its application across diverse situations, leading to a deep understanding and integration of knowledge. Moreover, Humans exploit curriculum learning (Bengio et al., 2009) to effectively learn new knowledge by progressing from easy to hard levels. In light of this, we combine them and design curricular meaningful learning to effectively remedy the diagnosed knowledge deficiencies of $\mathcal{L}$.

Table 1: Grouped knowledge deficiencies and the corresponding size of synthesized examples.

| Group | RE Interval | Synthesized Examples |
|---|---|---|
| **Easy** | $0.1 \leq \text{RE} < 0.4$ | 1 |
| **Normal** | $0.4 \leq \text{RE} < 0.7$ | 2 |
| **Hard** | $0.7 \leq \text{RE} < 1.0$ | 3 |
| **Unfair** | $\text{RE} \geq 1.0$ | 4 |

Firstly, we employ meaningful learning strategy to synthesize varying examples in diverse scenarios according to the deficiency severity. It is inspired by meaningful learning in humans and the insights that LLMs typically require more tokens or examples to learn the knowledge if they have less prior understanding of it (Ovadia et al., 2023; Gekhman et al., 2024). This strategy can reduce the cost and make deficiency remedy more efficient. Specifically, for the diagnosed deficiencies of $\mathcal{L}$, we divide them into 4 groups according to the severity (RE) of them. For each group, we heuristically assign a number, which indicates the number of diverse examples we should synthesize for each deficiency in the group. The detailed groups and assigned numbers are shown in Table 1. Subsequently, we adopt ChatGPT (Achiam et al., 2023) to synthesize the specified number of examples for the deficiencies in each group. The deficiency (knowledge and the corresponding query) is harnessed to guide the data synthesis process. Each synthesized example contains an input $X$ and an output $Y$.

Secondly, we devise a curricular remedy strategy to remedy the knowledge deficiencies in LLM $\mathcal{L}$ from minor to severe. Specifically, we sort the generated examples in ascending order based on the severity of their knowledge deficiencies, and then feed them into training $\mathcal{L}$ sequentially. For each example $< X, Y >$, we train $\mathcal{L}$ autoregressively to maximize a conditional probability:

$$
\mathcal{L}(X, Y, \theta) = -\sum_t log_{p_\theta}(Y_t | X, Y_{<t}),
\tag{4}
$$

where $\theta$ denotes parameters of $\mathcal{L}$. This approach results in an updated $\mathcal{L}$ with deficiencies remedied.

Table 2: Overall performance of LaMer and baselines. **Bold** numbers denote the best performance among all methods. *Average* denotes the performance averaged across all benchmarks.

| LLMs | Size | Methods | Comm. | AGIEval | ARC | MMLU | BBH | CRASS | GSM-Plus | *Average* |
|---|---|---|---|---|---|---|---|---|---|---|
| **Mistral** | 7B | Base | 67.56 | 32.82 | 74.15 | 49.75 | 28.47 | 71.67 | 29.91 | 50.62 |
| | | AugGPT | 74.53 | 33.03 | 76.93 | 52.56 | **33.82** | 81.67 | 13.78 | 52.33 |
| | | Naive | 67.42 | 33.60 | 72.66 | 50.58 | 32.22 | 80.00 | **31.19** | 52.52 |
| | | Single | 69.47 | **34.87** | 74.85 | 51.44 | 29.53 | 80.00 | 29.03 | 52.74 |
| | | LaMer (Ours) | **75.10** | 34.50 | **77.22** | **54.52** | 33.72 | **88.33** | 29.23 | **56.09** |
| **LLaMA-3** | 8B | Base | 74.84 | 39.45 | 86.29 | 59.72 | 39.85 | 76.67 | **61.93** | 62.82 |
| | | AugGPT | 76.75 | 38.13 | 86.04 | **60.29** | 36.60 | 76.67 | 14.92 | 55.63 |
| | | Naive | 71.28 | 33.86 | 84.50 | 57.86 | **40.93** | 83.33 | 61.56 | 61.90 |
| | | Single | 76.36 | 39.72 | 84.72 | 58.97 | 36.53 | 76.67 | 55.85 | 61.26 |
| | | LaMer (Ours) | **78.15** | **40.85** | **86.41** | 60.24 | 39.69 | **88.33** | 59.22 | **64.70** |
| **Qwen2** | 7B | Base | 71.88 | 42.10 | 85.48 | 59.53 | 37.58 | 85.00 | 61.79 | 63.34 |
| | | AugGPT | **79.41** | 42.71 | **89.78** | **63.66** | 40.00 | **90.00** | 14.75 | 60.04 |
| | | Naive | 75.62 | 42.80 | 88.40 | 62.50 | 39.34 | 86.67 | **61.82** | 65.31 |
| | | Single | 76.41 | 44.30 | 87.64 | 61.98 | 39.69 | 80.00 | 56.62 | 63.81 |
| | | LaMer (Ours) | 78.13 | **45.02** | 88.62 | 62.48 | **40.56** | 86.67 | 61.44 | **66.13** |
| **Gemma-1.1** | 2B | Base | 53.26 | 25.80 | **49.97** | 33.73 | 23.29 | 38.33 | **7.08** | 33.07 |
| | | AugGPT | 55.42 | 25.76 | 47.73 | 33.69 | 24.31 | 36.67 | 3.95 | 32.50 |
| | | Naive | 55.38 | 25.78 | 47.79 | 32.23 | 24.24 | 36.67 | 5.99 | 32.58 |
| | | Single | 52.85 | 24.51 | 46.67 | 34.00 | 25.03 | 31.67 | 5.86 | 31.51 |
| | | LaMer (Ours) | **55.81** | **25.81** | 48.91 | **34.40** | **25.26** | **41.67** | 6.69 | **34.08** |

## 3 EXPERIMENTS

### 3.1 INVESTIGATED LLMS

We adopt 4 open-source LLMs for experiments to illustrate the general applicability of our proposed LaMer: (1) **Mistral-7B-Instruct-v0.2** (Jiang et al., 2023) (Mistral) is an efficient chat LLM. (2) **LLaMA-3-8B-Instruct** (Meta, 2024) (LLaMA-3) is a dense LLM with massive pre-training. (3) **Qwen2-7B-Instruct** (Yang et al., 2024) (Qwen2) is a powerful multilingual LLM. (4) **Gemma-1.1-2B-IT** (Team et al., 2024) (Gemma-1.1) is a powerful small-scale chat LLM.

### 3.2 BASELINES

We adopt a wide range of baselines for comprehensive comparisons: (1) **Base** employs the base LLMs in Section 3.1 to answer questions in each benchmark. (2) **AugGPT** (Dai et al., 2023) uses ChatGPT (Achiam et al., 2023) to generate questions and answers to augment LLMs with SFT. AugGPT does not provide chain-of-thought (Wei et al., 2022) in generated examples. (3) **Naive** is an SFT method that randomly samples several pieces of knowledge from the knowledge base to synthesize new examples without considering whether the specific LLM possesses deficiencies on the sampled knowledge. (4) **Single** follows a similar process to LaMer but synthesizes only 1 example per knowledge deficiency. Hence, the training data of Single is 40% of LaMer or the other methods.

### 3.3 EVALUATION BENCHMARKS

We choose 7 **OOD** benchmarks ranging from reasoning to language understanding, to evaluate the performance of LaMer and baselines: (1) **Comm.** (Xiong et al., 2023) is a collection of 6 commonsense reasoning datasets. (2) **AGIEval** (Zhong et al., 2024) consists of diverse sets of standardized tests ranging from college admission tests (such as GRE and GMAT) to national civil service examinations. (3) **ARC** (Clark et al., 2018) is the AI2 Reasoning Challenge, which is a benchmark of science exams spanning Grade 3 to Grade 9 with easy (ARC-e) and challenge (ARC-c) subsets. (4) **MMLU** (Hendrycks et al., 2021) aims to evaluate language comprehension, knowledge, and reasoning skills of LLMs with 57 tasks. (5) **BBH** (Suzgun et al., 2023) is a subset of Big-Bench (Srivastava et al., 2023), which contains 23 hardest tasks focusing on challenging scenarios.

(6) **CRASS** (Frohberg & Binder, 2022) measures counterfactual reasoning in language models. (7) **GSM-Plus** (Li et al., 2024b) is a comprehensive math benchmark for evaluating the robustness of LLMs. We only keep the examples that possess valid answers for evaluation.

### 3.4 IMPLEMENTATION DETAILS

For knowledge extraction, we choose e-CARE (Du et al., 2022) and GSM8K (Cobbe et al., 2021), discarding the labels to obtain query set $D$. The size of GenericsKB after filtering is 200K. We utilize FlagEmbedding (Zhang et al., 2023a) with beg-large-en-v1.5 to encode facts and queries. We acquire $m = 4$ facts for each query. Since GSM8K is far from GenericsKB, we use ChatGPT to generate $m = 4$ pieces of knowledge for GSM8K. We generate $n = 2$ responses for each query.

For the label-free knowledge deficiency diagnosis step, the knowledge and corresponding query with an RE higher than $\tau = 0.1$ is treated as a knowledge deficiency of $\mathcal{L}$. The size of selected knowledge deficiencies can refer to Appendix E.

For the meaningful learning strategy, we utilize a prompt to instruct ChatGPT to synthesize examples. Finally, we synthesize 3,750 examples to enhance Mistral, Qwen2, and Gemma-1.1, while 1,250 examples are synthesized to enhance LLaMA-3 due to denser knowledge in it.

For the curricular deficiency remedy strategy, we adopt LoRA (Hu et al., 2022) for parameter-efficient fine-tuning. The rank $r$ and $\alpha$ of LoRA are 128 and 8, respectively. We train LaMer for 3 epochs with a learning rate of $5e$-5. The batch size is 32. The optimizer we used is Adam (Kingma & Ba, 2014). Two NVIDIA A100 80GB PCIe GPUs are used for training and the following evaluation.

For AugGPT and Naive, we randomly sample 3,750 facts from GenericsKB and the generated knowledge of GSM8K to generate the same number of training examples as LaMer for each LLM. While for Single, we randomly sample one example for each deficiency from the training data of LaMer. Therefore, Single enhances Mistral, Qwen2, and Gemma-1.1 with 1,500 examples, and it utilizes 600 examples to enhance LLaMA-3. The whole data synthesis process and the training setup are the same as LaMer. All prompts for the preprocess and data synthesis can refer to Appendix C. We utilize gpt-3.5-turbo-0125 for all relevant implementations based on ChatGPT.

### 3.5 EVALUATION DETAILS

For all benchmarks, we utilize free-form generation to evaluate all methods, the evaluation prompts for different LLMs can refer to Appendix D. The performance of each method on each benchmark is averaged across all tasks in the corresponding benchmark.

### 3.6 OVERALL RESULTS

The overall results are shown in Table 2, from which we can have the following observations:

(1) On average, LaMer outperforms all baselines across different LLMs, which is mainly due to the effectiveness of LaMer in diagnosing and remedying knowledge deficiency. This also reveals the general applicability of LaMer, making it a plug-and-play method to improve LLMs.

(2) Data augmentation methods can surpass the base LLM on most benchmarks, while Naive and Single show performance drops compared to LLaMA-3 and Gemma-1.1, as LLaMA-3 and Gemma-1.1 possess dense knowledge in its parameters. Naive and Single could supplement some knowledge to them but cause them to forget more useful knowledge.

(3) Interestingly, Single, which is trained with 40% training data of LaMer (one example for each knowledge deficiency), can achieve comparable performance with Naive on Mistral and LLaMA-3 with much fewer training data. This indicates Naive method can produce more redundant data that the base LLM already possesses, while detecting knowledge deficiencies in LLMs can apply targeted improvements, making it more efficient and less costly than other data synthesis methods.

(4) LaMer excels Single across all LLMs, this is because more severe knowledge deficiencies require more and diverse examples for LLMs to effectively remedy them (Gekhman et al., 2024). LaMer synthesizes more examples for knowledge deficiencies with higher severity, whereas Single generates

only one example for each knowledge deficiency. As a result, many knowledge deficiencies in the LLMs are not adequately remedied by Single.

(5) AugGPT achieves the worst results on GSM-Plus, this is because math problems demand multiple reasoning steps to solve. AugGPT directly offers the answer to the math problems, making it hard to develop precise calculations to arrive at the right answer. Furthermore, AugGPT can reach the best performance on several benchmarks (such as ARC on Qwen2), this is because problems in these benchmarks might be better answered with just a step, which is also indicated in Zhang et al. (2023b).

(6) Qwen2 could exceed LLaMA-3 across different methods with fewer parameters, we suppose this is mainly due to the massive post-training efforts on Qwen2.

## 4 CASE STUDY

Figure 3 shows a case to demonstrate how LaMer diagnoses and remedies a deficiency in Mistral: (a) For an query about "silver", we obtain the knowledge that "silver is a mildly toxic element". (b) The LLM offers a wrong response which produce a distribution $P = [0.63, 0.37]$ over the options. After providing the LLM with the knowledge, the LLM offers a new response with a different distribution $Q = [0.15, 0.85]$. The relative entropy between $P$ and $Q$ is 0.60, which means the knowledge brings a lot of information to the LLM. Hence, the combination of the knowledge and the query is a knowledge deficiency of the LLM. (c) According to Table 1, this knowledge deficiency is in normal group.

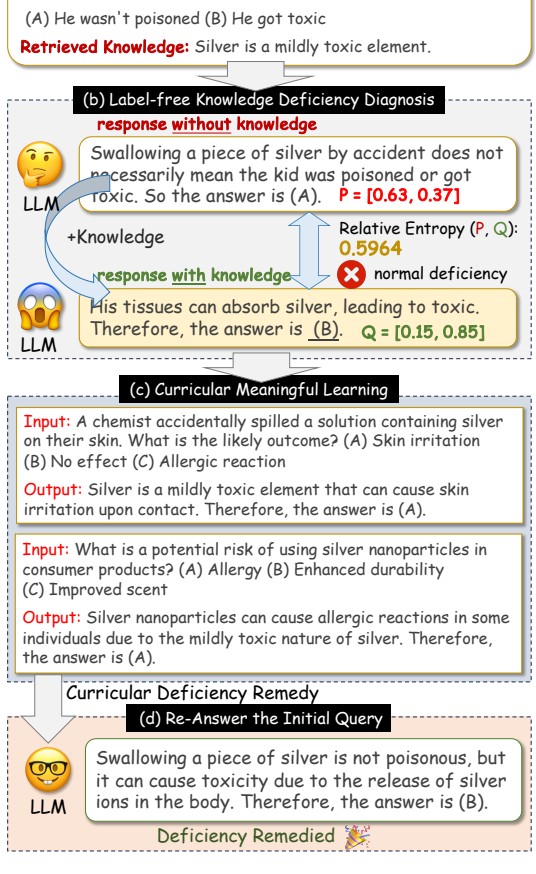

Figure 3: A case showing how LaMer diagnoses and remedies a knowledge deficiency in Mistral.

Thus, we ask ChatGPT to synthesize two examples for this deficiency. These two examples are two different applications of the knowledge in this deficiency. Next, the synthesized examples are used to train the LLM. (d) After curricular meaningful learning, Mistral offers correct answer. The deficiency is remedied.

## 5 FURTHER ANALYSIS

To further investigate the effectiveness of LaMer, we design several ablation studies and in-depth analyses: (1) comparisons between LaMer and label-reliant methods to demonstrate the efficiency and strengths of LaMer; (2) an effectiveness analysis of LaMer on remedying deficiencies by obtain the statistics on remedied examples of each method; (3) an ablation study to investigate the effect of curricular deficiency remedy strategy; (4) a significance analysis to inspect the roles of helpful and misleading situations in label-free knowledge deficiency diagnosis.

Table 3: Effectiveness of different methods on detecting difficiencies of Mistral based on e-CARE (Du et al., 2022) dataset.

| Methods | Label-free | P | R | F1 |
|---|---|---|---|---|
| Golden Label | No | 100 | 100 | 100 |
| Perplexity | No | 48.46 | 34.24 | 40.10 |
| Random | Yes | 35.23 | 35.23 | 35.23 |
| Relative Entropy | Yes | 40.34 | 64.30 | 49.58 |

## 5.1 COMPARISONS TO LABEL-RELIANT METHODS

We conduct an analysis to reveal the efficiency of different methods in diagnosing deficiencies based on Mistral-7B-Instruct-v0.2 (Jiang et al., 2023) and e-CARE (Du et al., 2022). Details can refer to Appendix F.1. Results are shown in Table 3, our proposed relative entropy method outperforms perplexity by recalling more deficiencies. This proves the feasibility of our proposed relative entropy method.

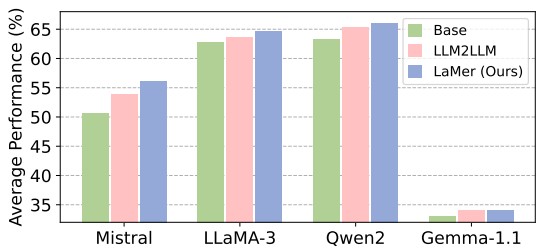

Figure 4: Average performance across all benchmarks of base LLMs, LLM2LLM, and LaMer.

Furthermore, we adopt a label-reliant method LLM2LLM for comparison. LLM2LLM (Lee et al., 2024) uses labeled data to identify erroneous examples in existing LLMs, and then synthesizes similar examples to improve a specific LLM. We use the labels of e-CARE and GSM8K to obtain 3,750 error examples, and then we generated similar examples based on the error examples. The number of training examples is the same as LaMer for each LLM. Results are shown in Figure 4 (full results can refer to Appendix F.2), we can find: LaMer has varying advantages over LLM2LLM across LLMs. Although LLM2LLM can precisely detect the errors of each LLM, the errors are limited to the given initial dataset. LaMer diagnoses knowledge deficiencies, which can improve the coverage of diagnosed deficiencies in each LLM.

## 5.2 EFFECTIVENESS IN REMEDYING DEFICIENCIES

We conduct an analysis to examine if LaMer can effectively remedy more error examples. Specifically for each method, we first tally the number of examples in each benchmark that base LLM is wrong but correct after the enhancement by each method (denote as remedied examples). After that, we normalize the values across different methods. Finally, we use spider charts to visualize them. We choose Mistral and LLaMA-3 for investigation. Results are in Figure 5:

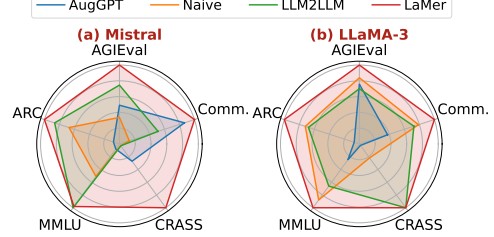

Figure 5: Normalized remedied examples on (a) Mistral and (b) LLaMA-3 across benchmarks.

(1) LaMer has advantages over baselines on all benchmarks, this is attributed to that LaMer can diagnose more deficiencies via relative entropy, and remedy them via curricular meaningful learning.

(2) In Table 2, although AugGPT can exceed LaMer on some benchmarks, it does not get the upper hand in Figure 5. This is mainly due to AugGPT cannot diagnosis the deficiencies of LLMs.

## 5.3 EFFECT OF CURRICULAR DEFICIENCY REMEDY

To investigate the effect of curricular deficiency remedy in LaMer, we randomly shuffle the data synthesized by LaMer to train the base LLMs (LaMer*). All the training and evaluation settings are the same as LaMer. We denote LaMer with shuffled data as LaMer*. The results are shown in Table 4 (full results can refer to Appendix F.3), we can observe:

(1) LaMer outperforms LaMer* regarding each LLM, since remedying less severe deficiencies helps remedy more severe ones. Curricular meaningful learning can make LLMs learn new knowledge more efficiently (Xiong et al., 2024).

Table 4: Ablation study on curricular deficiency remedy.

| LLMs | Size | Methods | *Average* |
|---|---|---|---|
| **Mistral** | 7B | LaMer | **56.09** |
| | | LaMer* | 55.64 |
| **LLaMA-3** | 8B | LaMer | **64.70** |
| | | LaMer* | 64.47 |
| **Qwen2** | 7B | LaMer | **66.13** |
| | | LaMer* | 65.50 |
| **Gemma-1.1** | 2B | LaMer | **34.08** |
| | | LaMer* | 33.80 |

(2) When switched to randomly shuffled data, LaMer* only suffers small performance drops. This claims that the performance of LaMer improvement is stable and LaMer has robust applicability.

(3) LaMer* achieves higher performance than baselines in Table 2 on Mistral, LLaMA-3, and Qwen2. It clarifies that the advantages of LaMer primarily stem from the deficiencies diagnosis process.

## 5.4 Helpful and Misleading Knowledge

The knowledge deficiencies can be caused by two kinds of knowledge: (1) helpful knowledge has a very positive impact on the correct answer; (2) misleading knowledge leads LLMs to choose the wrong answer with higher confidence than the right answer.

To find out the significance of the deficiencies caused by these two kinds of knowledge, we split the discovered knowledge deficiencies into two groups (Helpful and Misleading) with the help of golden labels. Then we respectively train the LLM with data synthesized based on Helpful and Misleading deficiencies. Finally, after evaluation, we respectively analyze the unique remedied examples brought by the data synthesized based on Helpful and Misleading deficiencies. Results are shown in Figure 6, we can have the following observations:

(1) Deficiencies caused by helpful and misleading knowledge are similarly significant. They can help remedy comparable yet distinct examples. Both highlight valid knowledge deficiencies.

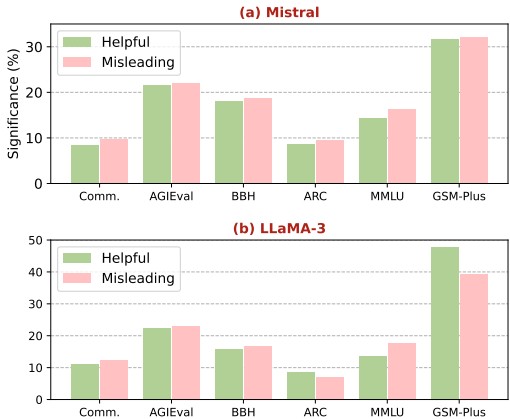

Figure 6: The portion of unique examples that are remedied based on the deficiencies caused by helpful and misleading knowledge on (a) Mistral and (b) LLaMA-3.

(2) Deficiencies caused by misleading knowledge have slightly greater significance than those caused by helpful knowledge, as they often have a larger impact, and is often more difficult and complex.

## 6 Related Work

### 6.1 Continual Training on LLMs

LLMs (Jiang et al., 2023; Meta, 2024) have demonstrated strong generalization capabilities, may not always perform well on general or specific individual capabilities. Recent works involved additional training to customize or enhance LLMs.

For enhancing the general capabilities of LLMs, Xie et al. (2023) and Li & Lee (2024) conducted continual pre-training on LLMs for alignment. Cui et al. (2023) and Huozi-Team (2024) conducted adaptions with massive Chinese data to enhance the Chinese capabilities of LLaMA (Touvron et al., 2023). Taori et al. (2023) utilized 52K instruction data from ChatGPT (Achiam et al., 2023) to obtain a strong instruction-following LLM. Vicuna (Chiang et al., 2023) adopted data from ShareGPT (ShareGPT, 2023) to train LLaMA and achieved 90% quality of ChatGPT. Orca (Mukherjee et al., 2023) and Orca-2 (Mitra et al., 2023) adopt progressive learning and more data from GPT-4 (Achiam et al., 2023) to further enhance the general abilities of LLMs. Different from the others, WizardLM (Xu et al., 2024) designed an evolve-instruct prompt to distill instruction data from ChatGPT with varying difficulty, encouraging LLMs to follow complex instructions. Zephyr (Tunstall et al., 2023) employed DPO (Rafailov et al., 2024) to align LLMs with human preference, while SPIN (Chen et al., 2024) devised a self-play method to achieve this. For enhancing specific abilities of LLMs, researchers used task-specific data to improve the desired abilities of LLMs such as math (Luo et al., 2023; Tang et al., 2024), code (Guo et al., 2024a), and reasoning (Ying et al., 2024).

Our work aims to detect the knowledge deficiencies of LLMs and apply appropriate remedies to repair them, which can serve as a complement to existing methods.

### 6.2 Evaluation of LLMs

Since LLMs have broad capabilities, evaluating LLMs becomes a tough and widely concerned issue. Some works constructed benchmarks or evaluation data to evaluate LLMs from general and specialized perspectives, such as natural language understanding (Hendrycks et al., 2021; Li et al.,

2023; Wang et al., 2024), reasoning (Frohberg & Binder, 2022; Suzgun et al., 2023; Zhong et al., 2024), math (Li et al., 2024b; Liu et al., 2024), coding (Peng et al., 2024; Zhang et al., 2024), etc. Different from using benchmarks with objective questions, some works started adopting LLMs to evaluate LLMs subjectively. Liu et al. (2023) applied GPT-4 as to assess the quality of generated text in various pesperctives (such as coherence) and achieved better alignment with human evaluators. Bai et al. (2024) evaluated the performance of existing LLMs with self-evaluation and peer-evaluation, achieving more precise judgments of existing LLMs. Furthermore, Zheng et al. (2024) devised a chat framework to evaluate LLMs based on the discussions among LLMs.

Our work emploies relative entropy to diagnose the knowledge deficiencies in LLMs based on a knowledge base. We also follow previous work to adopt objective benchmarks to evaluate LLMs.

## 7 CONCLUSION

In this paper, we design a label-free curricular meaningful learning framework (LaMer) based on relative entropy to first automatically discover the knowledge deficiencies from massive label-free user queries. Then we devise curricular meaningful learning which consists of a meaningful learning strategy and a curricular deficiency remedy strategy, to efficiently and effectively remedy the discovered knowledge deficiencies of corresponding LLM. The experiments show that our proposed LaMer can improve the coverage of diagnosed deficiencies, and surpass the baselines, making LLMs enhancement free of labeled data. The relative-entropy-based deficiency method provides a robust, efficient, and label-free deficiency diagnostic tool for existing LLMs to further unlock their potential.

## ACKNOWLEDGEMENT

The research in this article is supported by the New Generation Artificial Intelligence of China (2024YFE0203700), National Natural Science Foundation of China under Grants U22B2059, 62576124, and 62576102.

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

## A USE OF LLMs

We use ChatGPT to check the grammar and polish writing. We do not use LLMs for experiments or idea creating.

## B LIMITATIONS

This paper still have some limitations: (1) We require a related knowledge base for given queries. While for domain-specific queries, obtaining such a knowledge is difficult. (2) LaMer does not work well on math datasets. (3) Complex scenarios such complex reasoning could be further discussed.

## C   PROMPTS FOR IMPLEMENTATION

### C.1   PROMPT FOR PROCESS GSM8K

---

**Prompt for Processing GSM8K**

You are an expert in math problems.

First, please generate some background knowledge which can be used to solve this question. The knowledge SHOULD be general and applicable, Your generated knowledge CANNOT be question-specific. Like commonsense reasoning, I can give the knowledge "Acid is corrosive." Just list the knowledge.

Second, please answer this question by giving a short explanation and then give the answer. Third, please generate some distractors for your answer and index them like "(A)__ (B)__ ···".

The answer and distractors should be as concise as possible. Your response SHOULD follow the following format: Background Knowledge: [The generated knowledge] Explanation: [Steps to achieve the answer] Answer: [A pure math part] Distractors: [Wrong answers]

---

Since GSM8K (Cobbe et al., 2021) contains math problems and without options, we adopt ChatGPT to generate distract options and knowledge for each query in GSM8K. The following is the prompt:

## C.2 PROMPT FOR SYNTHESIZING DATA

---

**LaMer: e-CARE**

You are an expert in creating reasoning and language understanding questions.

Here are the requirements:
1. You are given a fact and some examples for reference, the fact can implicitly guide the solution for the reference examples. You should understand the Internal Mechanism of this guidance to make new examples.
2. Each created example should contain a question, several options ($\geq$3), an answer, and a short explanation for the answer.
3. The options should be a short sentence or phrase rather than a single word whenever possible.
4. Don't make any commonsense mistakes and ensure that your solutions are accurate.
5. You SHOULD propose totally new reasoning questions in various areas, including but not limited to history, law, medical, math, science, computer science, psychology, AI, politics, economics, etc.
6. NOTE that the fact should be an implicit explanation for obtaining the true answer, which means the fact SHOULD NOT appear explicitly in the questions or the options.
7. Only one option is the correct answer, the other options should be much less plausible than the correct option or they are just wrong options.
8. Therefore is no explanation in the reference examples, you SHOULD generate an explanation first and then give the answer for your generated new questions.
9. The question could be in any form, such as "Why, What, How, Which" etc. You can also add a premise to form a question.
10. The created examples cannot be different just in nouns.

Reference:
Knowledge: {      }
Examples: {      }

You MUST generate {     } new examples. The examples MUST be totally different from each other and the reference examples. Please return your response in the form:
Question: [QUESTION]
Options: [CANDIDATE OPTIONS]
Answer: [The option index of the answer such as (B)]
Explanation: [A concise explanation for the answer]

Question: [QUESTION]
Options: [CANDIDATE OPTIONS]
Answer: [The option index of the answer such as (B)]
Explanation: [A concise explanation for the answer]

......

---

---

**LaMer: GSM8K**

You are an expert in creating math questions. Your goal is to generate a set of new math questions based on the reference fact and examples.

Here are the requirements:
1. The example should contain a question, a solution to derive the answer to the question, several options ($¿$=3), and an answer.
2. Don't make any commonsense mistakes and ensure that your solutions are accurate.
3. You SHOULD propose totally new reasoning questions.
4. NOTE that the fact should be an implicit guidance for obtaining the true answer, which means the fact SHOULD NOT appear explicitly in the questions, solutions, or options.
5. Only one option is the correct answer.
6. There are no solutions in the reference examples, you SHOULD generate solutions first and then give the answer for your generated new questions.
7. The question could be in any form. You can also add a premise to form a question.

Reference:
Knowledge: {        }
Examples: {       }

You MUST generate {      } new examples. The examples MUST be totally different from each other and the reference examples. Please return your responses in the form:
Question: [QUESTION]
Solution: [A CONCISE step-by-step SOLUTION to DERIVE the ANSWER to the Question]
Options: [CANDIDATE OPTIONS containing the answer]
Answer: [The option of the answer such as (B) $15]

Question: [QUESTION]
Solution: [A CONCISE step-by-step SOLUTION to DERIVE the ANSWER to the Question]
Options: [CANDIDATE OPTIONS containing the answer]
Answer: [The option of the answer such as (B) $15]

......

---

## D   PROMPT FOR EVALUATION

---

**Mistral**

[INST]Question: {      }
Options: {      }[/INST]
......

---

**LLaMA-3 and Qwen2**

user
Question: {      }
Options: {      }

assistant
......

---

Table 5: Overall performance of LaMer, LLM2LLM, and baselines. **Bold** numbers denote the best performance among all methods. *Average* denotes the average performance across all benchmarks.

| LLMs | Size | Methods | Label-free | Comm. | AGIEval | ARC | MMLU | BBH | CRASS | GSM-Plus | *Average* |
|---|---|---|---|---|---|---|---|---|---|---|---|
| **Mistral** | 7B | Base | - | 67.56 | 32.82 | 74.15 | 49.75 | 28.47 | 71.67 | 29.91 | 50.62 |
| | | AugGPT | Yes | 74.53 | 33.03 | 76.93 | 52.56 | 33.82 | 81.67 | 13.78 | 52.33 |
| | | Naive | Yes | 67.42 | 33.60 | 72.66 | 50.58 | 32.22 | 80.00 | 31.19 | 52.52 |
| | | Single | Yes | 69.47 | 34.87 | 74.85 | 51.44 | 29.53 | 80.00 | 29.03 | 52.74 |
| | | LLM2LLM | No | 74.59 | 34.44 | 78.66 | 55.03 | 34.58 | 83.33 | 16.33 | 54.00 |
| | | LaMer (Ours) | Yes | 75.10 | 34.50 | 77.22 | 54.52 | 33.72 | 88.33 | 29.23 | **56.09** |
| **LLaMA-3** | 8B | Base | - | 74.84 | 39.45 | 86.29 | 59.72 | 39.85 | 76.67 | 61.93 | 62.82 |
| | | AugGPT | Yes | 76.75 | 38.13 | 86.04 | 60.29 | 36.60 | 76.67 | 14.92 | 55.63 |
| | | Naive | Yes | 71.28 | 33.86 | 84.50 | 57.86 | 40.93 | 83.33 | 61.56 | 61.90 |
| | | Single | Yes | 76.36 | 39.72 | 84.72 | 58.97 | 36.53 | 76.67 | 55.85 | 61.26 |
| | | LLM2LLM | No | 75.86 | 36.92 | 87.36 | 60.24 | 39.95 | 88.33 | 56.58 | 63.61 |
| | | LaMer (Ours) | Yes | 78.15 | 40.85 | 86.41 | 60.24 | 39.69 | 88.33 | 59.22 | **64.70** |
| **Qwen2** | 7B | Base | - | 71.88 | 42.10 | 85.48 | 59.53 | 37.58 | 85.00 | 61.79 | 63.34 |
| | | AugGPT | Yes | 79.41 | 42.71 | 89.78 | 63.66 | 40.00 | 90.00 | 14.75 | 60.04 |
| | | Naive | Yes | 75.62 | 42.80 | 88.40 | 62.50 | 39.34 | 86.67 | 61.82 | 65.31 |
| | | Single | Yes | 76.41 | 44.30 | 87.64 | 61.98 | 39.69 | 80.00 | 56.62 | 63.81 |
| | | LLM2LLM | No | 80.29 | 43.74 | 89.72 | 63.21 | 40.88 | 80.00 | 60.09 | 65.42 |
| | | LaMer (Ours) | Yes | 78.13 | 45.02 | 88.62 | 62.48 | 40.56 | 86.67 | 61.44 | **66.13** |
| **Gemma-1.1** | 2B | Base | - | 53.26 | 25.80 | 49.97 | 33.73 | 23.29 | 38.33 | 7.08 | 33.07 |
| | | AugGPT | Yes | 55.42 | 25.76 | 47.73 | 33.69 | 24.31 | 36.67 | 3.95 | 32.50 |
| | | Naive | Yes | 55.38 | 25.78 | 47.79 | 32.23 | 24.24 | 36.67 | 5.99 | 32.58 |
| | | Single | Yes | 52.85 | 24.51 | 46.67 | 34.00 | 25.03 | 31.67 | 5.86 | 31.51 |
| | | LLM2LLM | No | 54.03 | 25.56 | 49.50 | 35.77 | 25.60 | 40.00 | 7.81 | 34.04 |
| | | LaMer (Ours) | Yes | 55.81 | 25.81 | 48.91 | 34.40 | 25.26 | 41.67 | 6.69 | **34.08** |

Table 6: The size of selected knowledge deficiencies.

| LLMs | Easy | Normal | Hard | Unfair |
|---|---|---|---|---|
| **Mistral** | 375 | 375 | 375 | 375 |
| **LLaMA-3** | 125 | 125 | 125 | 125 |
| **Qwen2** | 375 | 375 | 375 | 375 |
| **Gemma-1.1** | 375 | 375 | 375 | 375 |

```
Gemma-1.1

user
Question: {      }
Options: {      }
model
......
```

# E    SIZE OF SETECTED KNOWLEDGE DEFICIENCIES

The size of selected knowledge deficiencies of each LLM in each group can refer to Table 6.

Table 7: Statistics of e-CARE

| Dataset | Train | Dev | Test | Total |
|---|---|---|---|---|
| e-CARE | 14,928 | 2,132 | 4,264 | 21,324 |

Table 8: Ablation study results of the effect of the curricular deficiency remedy, LaMer* denotes that we train corresponding LLMs with randomly shuffled data. Bold numbers denote better performance between LaMer and LaMer*.

| LLMs | Size | Methods | Comm. | AGIEval | ARC | MMLU | BBH | CRASS | GSM-Plus | *Average* |
|---|---|---|---|---|---|---|---|---|---|---|
| **Mistral** | 7B | LaMer | **75.10** | **34.50** | 77.22 | **54.52** | **33.72** | **88.33** | **29.23** | **56.09** |
| | | LaMer* | 74.90 | 34.07 | **77.48** | 54.30 | 33.66 | 86.67 | 28.42 | 55.64 |
| **LLaMA-3** | 8B | LaMer | **78.15** | 40.85 | **86.41** | **60.24** | **39.69** | **88.33** | 59.22 | **64.70** |
| | | LaMer* | 77.67 | **40.93** | 86.28 | 60.07 | 38.47 | **88.33** | **59.55** | 64.47 |
| **Qwen2** | 7B | LaMer | **78.13** | **45.02** | **88.62** | **62.48** | **40.56** | **86.67** | **61.44** | **66.13** |
| | | LaMer* | 77.51 | 44.33 | 88.36 | 62.12 | 39.80 | 85.00 | 61.36 | 65.50 |
| **Gemma-1.1** | 2B | LaMer | **55.81** | **25.81** | **48.91** | **34.40** | **25.26** | **41.67** | **6.69** | **34.08** |
| | | LaMer* | 55.05 | 25.32 | 48.63 | 34.16 | 25.18 | **41.67** | 6.60 | 33.80 |

# F  FURTHER ANALYSIS

## F.1  DEFICIENCY DIAGNOSIS

We formally describe the baselines as follows:

- **Golden Label** adopt the labels to judge the response of a specific LLM, this LLM would answer each question in a chain-of-thought (Wei et al., 2022) way. If the LLM gives the wrong answers according to the labels on some examples, then the examples are treated as the deficiencies of this LLM. We treat this method as the golden standard for diagnosing the knowledge deficiencies of LLMs.

- **Perplexity** computes the perplexity of each option based on a specific LLM, the option with the lowest perplexity is treated as the answer of the LLM, and then labels are introduced the judge the correctness of the LLM. Similar to Golden Label, wrongly answered examples are treated as the deficiencies of this LLM.

- **Random** method randomly samples the examples from a dataset.

- **Relative Entropy** is the method proposed in this paper.

We choose e-CARE (Du et al., 2022) as the dataset for experiments, which is an explainable causal reasoning dataset with two options in each example. The whole set of e-CARE (train, dev, and test) is adopted for experiments. Statistics of e-CARE can refer to Table 7. The LLM we used is Mistral-7B-Instruct-v0.2 (Jiang et al., 2023).

## F.2  FULL RESULTS OF LLM2LLM

We provide the full results of LaMer, all baselines, and LLM2LLM in Table 5, which demonstrates the advantages of our proposed LaMer.

## F.3  FULL RESULTS OF THE EFFECT OF CURRICULAR DEFICIENCY REMEDY

We provide the full results of the effect of curricular deficiency remedy Table 8, which demonstrates the curricular deficiency remedy strategy.

## F.4  VISUALIZATION OF GENERATED EXAMPLES

## F.5  CAUSES FOR THE ADVANTAGES OF LAMER

To explore the potential causes for the advantages of LaMer, we visualize the data synthesized by baselines and LaMer into 2D space. Specifically, we utilize FlagEmbedding (Zhang et al., 2023a) to represent each example in the synthesized data, and then reduce the dimensionality to 2 with the help of t-SNE (Van der Maaten & Hinton, 2008). Finally, we employ DBSCAN (Ester et al., 1996)

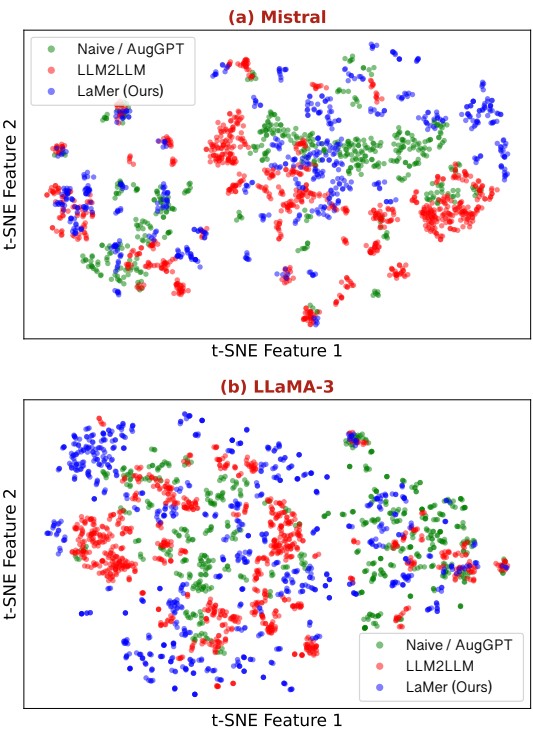

Figure 7: The distribution of synthesized data by baselines and LaMer on (a) Mistral and (b) LLaMA-3.

to discover clusters and remove the noise data points. The eps and min samples of DBSCAN are 1.5 and 3, respectively. We also adopt Mistral and LLaMA-3 for experiments. The visualization is shown in Figure 7, from which we can infer that, data synthesized by LaMer has a significantly higher proportion at the outer edges of the whole distribution (LaMer and baselines). This might be a potential cause for the advancement of LaMer, since LaMer can utilize relative entropy to effectively discover more deficiencies.

