# OpenReview forum: "Diagnosing and Remedying Knowledge Deficiencies in LLMs via Label-free Curricular Meaningful Learning"
_ICLR.cc/2026/Conference — ICLR 2026 Poster_

### Official Review · Reviewer_oqoY · 2025-10-23

**Soundness:** 3
**Presentation:** 3
**Contribution:** 3
**Rating:** 8
**Confidence:** 3

**Summary:**

The paper proposes LaMer, a label-free pipeline that (1) retrieves generic facts for each unlabeled query from an external knowledge base, (2) diagnoses “knowledge deficiencies” by comparing a model’s predictions with and without the retrieved fact, and (3) remedies those deficiencies by synthesizing training examples whose quantity scales with the diagnosed severity and training in an easy-to-hard curriculum. Using open-weight models, the method reports consistent gains over several augmentation baselines on seven out-of-distribution reasoning benchmarks, often matching or surpassing baselines with about 40% of the training data.

**Strengths:**

- The paper ties together retrieval, a simple distribution-shift diagnostic, and a curriculum that scales data to severity, turning unlabeled user queries into targeted training data without manual labels. The case study and “remedied examples” analysis make the mechanism concrete and inspectable.
- The paper contrasts label-free deficiency detection against perplexity and a label-reliant data-mining baseline, studies the curriculum order (vs. shuffling), and separates “helpful” vs. “misleading” retrieved facts—showing both expose repairable deficiencies.
- It specifies the KB (GenericsKB with confidence filtering), retrieval embedding (FlagEmbedding), the number of matched facts per query, the synthesis protocol, and PEFT training settings, which lowers the barrier to trying LaMer in practice.

**Weaknesses:**

- The diagnosis hinges on retrieved facts being relevant and on a chosen threshold to flag deficiencies. If retrieval drifts (spurious or overly generic facts) or if the threshold is mis-set, the pipeline may teach to noise.
- The severity buckets and the number of synthesized examples per bucket are fixed heuristics. While effective, it maybe interesting to explore the direction of an adaptive curricula (e.g., stopping rules per deficiency) or data-budget trade-offs across tasks and models.
- While “remedied examples” are counted, there’s no probing of whether the method inadvertently forgets useful adjacent knowledge or whether it changes calibration/uncertainty properties.

**Questions:**

- I am a bit curious about the retrieval robustness, how sensitive is the deficiency diagnosis to the retrieval pipeline (embedding model choice, number of facts, KB domain coverage)? What happens if retrieved facts are partially wrong or overly generic?
- How did you select the deficiency threshold, and how stable are results across plausible ranges? Could you replace hard thresholding with an adaptive percentile per domain or model?
- How would you adapt LaMer when queries are private or streaming (e.g., enterprise chat), and when external KBs are domain-specific or scarce?

---

> ### Author Response · Authors · 2025-11-21
>
> We would like to sincerely thank you for the careful reading of our manuscript and for the constructive comments.
> ## Weaknesses 1
> We have replaced the knowledge retrieval process with a random sampling process. For each query, we perform uniform sampling over the knowledge base to obtain four pieces of knowledge. After that, we follow the deficiency diagnosis step to compute RE. The following table shows the distribution of the RE in our method and the random sampling method.
>
> |Method|Easy|Normal|Hard|Unfair|
> | ----------------------- | :---: | :----: | :--: | :----: |
> |**Random**|3,182|937|322|556   |
> |**Ours**|10,973|3,683|1,971|5,926|
> |**[Random/Ours] Ratio**|0.29|0.25|0.16|0.09|
>
> We can find: (1) the retrieval noise does not impact too much on RE computation. (2) The higher the RE , the smaller the impact of noise on RE. This demonstrates that our method is robust to retrieval noise.
>
> ## Weaknesses 2
> Thanks to your constructive advice, an adaptive strategy of each deficiency and data-budget trade-offs across tasks as well as models is worth exploring for fine-grained improvement.  We are willing to deeply investigate these aspects in the future.
>
> ## Weaknesses 3
> Thanks to your constructive advice, injecting new knowledge while keeping useful adjacent knowledge is the research point of model editing, which is interesting to explore, and we are investigating this in our next work.
>
> ## Question 1
> A better retrieval result is the research point of retrieval, which requires another paper to investigate. While we aim to diagnose and remedy deficiencies of LLMs, we can use a web search engine to ensure the coverage.
>
> Additionally, we filter out the knowledge with confidence lower than 0.7, which can ensure the quality of the retrieved knowledge.
> ## Question 2
> We start from 0.1 because RE can achieve the best F1 in Table 3, and we choose 0.4, 0.7, and 1.0 to achieve uniform distribution. Furthermore, we have investigated the sensitivity of the threshold by designing two new grouping methods: (1) **Value-Based**: Easy [0.1-0.3), Normal [0.3, 0.6), Hard [0.6, 0.9), Unfair >= 0.9. (2) **Quantile-Based**: Use the quantiles (25% / 50% / 75%) of RE >= 0.1 in the diagnosis dataset as the boundaries to divide Easy, Normal, Hard, and Unfair. Finally, we follow LaMer (our method) to sample the deficiencies from each group and synthesize data to enhance Mistral, we only test these methods on Comm., AGIEval, and MMLU due to time limit. The following table shows the overall results:
> |Method|Comm.|AGIEval|MMLU|Average|
> | ------------------------------- | :---: | :-----: | :---: | :-----: |
> |Mistral (Base)|67.56|32.82|49.75|50.04|
> |LaMer with Value-based Grouping|75.26|35.24|54.14|54.88|
> |LaMer Quantile-Based Grouping|74.94|34.93|53.73|54.53|
> |LaMer (Ours)|75.10|34.50|54.52|54.71|
>
> We can find that LaMer can achieve stable performance with different grouping methods, which demonstrates the robustness of our proposed method on threshold sensitivity.
>
> Additionally, we use this fixed threshold strategy because: (1) fixed thresholds are easy to apply to all situations; (2) adaptive thresholds can be biased based on limited examples in all domains and models. Considering Occam's Razor, we choose the fixed one, and we are willing to explore an adaptive strategy to further improve the LLMs in the future.
>
> Our method can outperform baselines on 4 LLMs across 7 OOD benchmarks, which can demonstrate the stability and robustness of the fixed threshold.
> ## Question 3
> Thanks for your insightful advice.
>
> For private and streaming settings: (1) In an enterprise chat setting, LaMer can be deployed inside the organization’s boundary and operate on internal logs and internal KBs only. No queries or predictions need to leave the private environment. (2) For streaming queries, LaMer can be run in an online fashion: we maintain running statistics of deficiencies over a sliding window of recent queries, periodically remedying the discovered deficiency. The algorithm itself does not rely on access to the full data.
>
> When external KBs are domain-specific and scarce: (1) In vertical domains, the “external KB” naturally corresponds to internal, domain-specific resources (databases, manuals, FAQs, API docs, etc.). LaMer can directly use knowledge sampled from such internal sources, as we already do from our database in the experiments. (2) When the KB is small or sparse, LaMer still applies: it simply diagnoses deficiencies on the subset of queries that can be grounded to available knowledge. In other words, the coverage may decrease, but the methodology remains valid. (3) Moreover, small domain KBs can be amplified by simple bootstrapping strategies (e.g., templating over structured records, or letting an LLM paraphrase and slightly vary existing facts while keeping them consistent with the KB). LaMer does not require exhaustively complete knowledge; it only needs a set of trusted knowledge snippets to probe and remediate specific deficiencies.

---

### Official Review · Reviewer_4XWw · 2025-10-25

**Soundness:** 3
**Presentation:** 2
**Contribution:** 3
**Rating:** 6
**Confidence:** 4

**Summary:**

This paper introduces LaMer, a novel framework for diagnosing and remedying knowledge deficiencies in LLM without relying on labeled data. The core idea is to use relative entropy to identify knowledge gaps by measuring the change in an LLM's output distribution before and after being provided with relevant external knowledge facts. Based on the severity of the diagnosed deficiency, LaMer employs a curricular learning strategy to synthesize a varying number of diverse examples to progressively fine-tune the model. The empirical results are strong and consistent across four different LLMs and seven OOD benchmarks, demonstrating the effectiveness and efficiency of the proposed method.

**Strengths:**

1. The primary contribution (using relative entropy for label-free diagnosis of knowledge deficiencies) is well-motivated and novel. It addresses a critical and practical challenge in the continuous improvement of LLMs: how to perform targeted enhancements without expensive human annotation.
2. The integration of curricular and meaningful learning is well-executed. The paper clearly demonstrates that diagnosing deficiencies first and then applying a targeted, progressive remedy (from easy to hard) is more effective than naive data augmentation or random-order training.

**Weaknesses:**

1. The framework's effectiveness is highly dependent on two external components: a comprehensive knowledge base (GenericsKB) and a powerful teacher model (ChatGPT) for data synthesis. This reliance may limit its applicability in scenarios where a high-quality KB is unavailable or the cost of using a powerful synthesis model is prohibitive. The paper does not sufficiently discuss the impact of these dependencies.
2. The RE thresholds used to categorize deficiencies into "Easy," "Normal," "Hard," and "Unfair" are presented as heuristics. The paper lacks a rigorous justification or a sensitivity analysis for these values, making it unclear how robust the method is to these choices.
3.  The diagnosis step relies on knowledge retrieved via embedding similarity, which can be noisy and sometimes irrelevant. The paper does not address how such noisy knowledge might affect the stability of the posterior distribution Q and the resulting RE calculation. This raises concerns about whether the RE score is always a reliable indicator of a true knowledge deficiency.
4. The methodology description contains ambiguities that hinder clarity. For instance, the paper refers to the "negative log-likelihood (NLL) of each response oi conditioned on x,"(Section 2.2) but 'x' is not defined in the context.

**Questions:**

1. Regarding the knowledge retrieval process, how do you ensure the correctness of the knowledge recalled from the KB? Given that embedding-based matching can introduce noise, have you conducted any analysis on the impact of incorrectly retrieved or irrelevant knowledge on the RE calculation? How robust is the diagnosis mechanism to this noise?
2. In the Section 2.2, you refer to "The first situation suggests L might not grasp this knowledge or cannot properly apply this knowledge to problem-solving, while the second situation indicates that L does understand this knowledge but is easily misled by it.", providing a specific interpretation for two scenarios of knowledge impact. Can you provide the reasoning behind the claim that when knowledge has a negative impact (misleading), it indicates "L does understand this knowledge but is easily misled by it"? Why does this scenario not also suggest a failure to properly integrate or contextualize new information, which could be seen as a form of "not grasping" the knowledge in a given context?
3. Can you elaborate on the sensitivity of LaMer to the quality of its external components? Specifically, how would performance degrade if a less comprehensive knowledge base were used, or if a weaker, open-source model was used for data synthesis instead of ChatGPT?
4. Regarding the RE thresholds in Table 1, have you performed any experiments to analyze their sensitivity? How were these specific values (0.1, 0.4, 0.7, 1.0) determined, and how critical are they to the overall performance of the curricular remedy strategy?

---

> ### Author Response · Authors · 2025-11-21
>
> We would like to sincerely thank you for the careful reading of our manuscript and for the constructive comments.
> ## Weaknesses 1
> Thanks for your insightful suggestion. We will discuss this in the final version due to the page limit.
>
> We suppose an arbitrary knowledge base can work well in our method. If we cannot find a proper KB, then use the web searcher to obtain the knowledge. Additionally, we follow conventional methods to use strong LLMs (ChatGPT) to enhance smaller ones [1][2]. ChatGPT is not the only option, we can use Qwen3 or DeepSeek-R1 to play the role of ChatGPT. The following shows the results of LaMer enhanced with data synthesized by Qwen3-32B, which demonstrates the robustness of our method with open-source LLMs.
> | Method                          | Comm. | AGIEval |  ARC  | MMLU  |  BBH  | CRASS | GSM-Plus | Average |
> | :------------------------------ | :---: | :-----: | :---: | :---: | :---: | :---: | :------: | :-----: |
> | LaMer (Mistral)  with GPT-3.5   | 75.10 |  34.50  | 77.22 | 54.52 | 33.72 | 88.33 |  29.23   |  56.09  |
> | LaMer (Mistral)  with Qwen3-32B | 74.97 |  35.20  | 77.20 | 53.69 | 34.16 | 86.67 |  30.95   |  **56.12**  |
>
> [1] Gemma: Open Models Based on Gemini Research and Technology
>
> [2] QWEN2 TECHNICAL REPORT
> ## Weaknesses 2
> Thanks for your advice.
> With this categorization, our method can outperform baselines on 4 LLMs across 7 OOD benchmarks, which demonstrates that this categorization can bring stable improvement to LLMs. Furthermore, the search space is infinite, a heuristically chosen categorization can bring improvement, which indicates our method can work without careful design. We acknowledge that a more fine-grained selection of severity thresholds and example counts could potentially lead to further improvements, and we plan to explore this in future work.
> ## Weaknesses 3
> On the one hand, we aim to diagnose and remedy deficiencies in LLMs in a label-free setting, and retrieving precise knowledge is not the research point of our paper, which requires another paper to investigate. A better retriever can also improve our method. On the other hand, the RE score is sufficiently reliable to indicate a true deficiency and enhance LLMs based on the overall results in Table 2.
> ## Weaknesses 4
> Sorry for the ambiguities. We have checked the methodology description to fix all these issues.
> ## Question 1
> The knowledge with confidence lower than 0.7 (provided in GenericsKB) will be filtered out to ensure the quality. We aim to diagnose and remedy deficiencies in LLMs in a label-free setting, and retrieving precise knowledge is not the research point of our paper, which requires another paper to investigate.
>
> Additionally, our method can work on 4 LLMs across 7 OOD benchmarks with the embedding retriever, which, to some extent, reflects the robustness of our method.
>
> We acknowledge the advice of the reviewer, and we are willing to deeply investigate this in the future.
> ## Question 2
> We suppose that if LLMs do not understand the knowledge, then they will not be misled by it, this is why we describe the second situation in this way. If LLMs understand the knowledge and fail to properly integrate or contextualize new information, it is a case of “LLMs cannot properly apply the knowledge in problem-solving”.
> ## Question 3
> Please refer to the experiment in reponse to the weaknesses 1.
> ## Question 4
> We start from 0.1 because RE can achieve the best F1 in Table 3, and we choose 0.4, 0.7, and 1.0 to achieve uniform distribution. Due to time constraints, we are willing to investigate the sensitivity of the thresholds to divide groups in the final version.

---

> > ### Comment · Reviewer_4XWw · 2025-11-26
> >
> > I thank the authors for their response and for testing the framework with an open-source model (Qwen3-32B).
> > However, I am disappointed by the response regarding the core robustness issues:
> > 1.  **Retrieval Noise (W3/Q1):** The authors dismissed the concern about how noisy retrieval affects the RE metric by stating "retrieving precise knowledge is not the research point" and "requires another paper." I strongly disagree. Since the paper's main contribution is a *diagnosis framework* based on these retrieved signals, analyzing its sensitivity to noise is fundamental to the method's soundness, not out of scope.
> > 2.  **Threshold Sensitivity (W2/Q4):** The authors acknowledged that the thresholds were chosen heuristically to achieve a uniform distribution but failed to provide the requested sensitivity analysis. This leaves the robustness of the proposed method unverified.
> >
> > Given that these critical questions regarding the method's reliability and robustness remain unaddressed, I cannot raise my score and maintain my current rating.

---

> ### Author Response · Authors · 2025-12-01
>
> We thank the reviewer for clarifying the concerns, and we have conducted two extra experiments to demonstrate: (1) the impact of retrieval noise on RE computation, and (2) the sensitivity of RE threshold. We hope these can address the concerns of the reviewer.
>
> &nbsp;
>
> ### **1. Retrieval Noise (W3 & Q1)**
> We have replaced the knowledge retrieval process with a random sampling process. For each query, we perform uniform sampling over the knowledge base to obtain four pieces of knowledge. After that, we follow the deficiency diagnosis step to compute RE. The following table shows the distribution of the RE in our method and the random sampling method.
> | Method                  | Easy  | Normal | Hard | Unfair |
> | ----------------------- | :---: | :----: | :--: | :----: |
> | **Random**              | 3,182  |  937   | 322  |  556   |
> | **Ours**                | 10,973 |  3,683  | 1,971 |  5,926  |
> | **[Random/Ours] Ratio** | 0.29  |  0.25  | 0.16 |  0.09  |
>
> We can find: **(1) the retrieval noise does not impact too much on RE computation. (2) The higher the RE , the smaller the impact of noise on RE.** This demonstrates that our method is robust to retrieval noise.
>
> &nbsp;
>
>
> ### **2. Threshold Sensitivity (W2/Q4)**
> Additionally, we have investigated the sensitivity of the threshold by designing two new grouping methods: (1) **Value-Based**: Easy [0.1-0.3), Normal [0.3, 0.6), Hard [0.6, 0.9), Unfair >= 0.9. (2) **Quantile-Based**: Use the quantiles (25% / 50% / 75%) of RE >= 0.1 in the diagnosis dataset as the boundaries to divide Easy, Normal, Hard, and Unfair. Finally, we follow LaMer (our method) to sample the deficiencies from each group and synthesize data to enhance Mistral, we only test these methods on Comm., AGIEval, and MMLU due to time limit. The following table shows the overall results:
> | Method                          | Comm. | AGIEval | MMLU  | Average |
> | ------------------------------- | :---: | :-----: | :---: | :-----: |
> | Mistral (Base)                            | 67.56 |  32.82  | 49.75 |  50.04  |
> | LaMer with Value-based Grouping | 75.26 |  35.24  | 54.14 |  54.88  |
> | LaMer Quantile-Based Grouping   | 74.94 |  34.93  | 53.73 |  54.53  |
> | LaMer (Ours)                          | 75.10 |  34.50  | 54.52 |  54.71  |
>
> We can find that LaMer can achieve stable performance with different grouping methods, which demonstrates the robustness of our proposed method on threshold sensitivity.
>
> We hope these analyses can address the concerns of the reviewer.

---

### Official Review · Reviewer_85W6 · 2025-10-31

**Soundness:** 3
**Presentation:** 3
**Contribution:** 3
**Rating:** 6
**Confidence:** 4

**Summary:**

They propose a method to first find the knowledge deficiency in LLMs which, based on the literature, is the source of mistakes in reasoning. To do so they use user queries and using an external knowledge base, they get the relevant info for each query. Then using relative entropy, they detect the knowledge deficiency. Afterwards, using curricular meaningful learning, they propose a method to remedy this knowledge deficiency. This method is label free and does not require human supervision.
They show that this method achieves comparable results to baselines with only 40% training data.

**Strengths:**

No need for human annotation or labels.
Rich experiments
Clearly written and easy to understand

**Weaknesses:**

This method is limited to adding knowledge from GenericsKB.
They do not show how much of knowledge in GenericsKB are actually missing in the LLM they study (maybe for some of them the knowledge is already there but fine tuning only makes that knowledge sharp.)
Heavy reliance on ChatGPT (if ChatGPT makes errors, their method will as well).

**Questions:**

1. in the paper you mention that: "Subsequently, we adopt ChatGPT (Achiam et al., 2023) to synthesize the specified number of examples for the deficiencies in each group.". How do you make sure that chatGPT has enough and correct information for that knowledge?
2. why in Gemma-1.1 your method beats others in most of the cases but with Qwen2, it does not?
3. "We only keep the examples that possess valid answers for evaluation." what percentage of answers did you throw away? and how do you define “valid” here?
4. "we use ChatGPT to generate m= 4 pieces of knowledge for GSM8K". What if ChatGPT makes a mistake? how do you guarantee ChatGPT is correct?
5. Are you sure numbers is Table 6 are correct?
6. "Finally, we synthesize 3,750 examples to enhance Mistral, Qwen2, and Gemma-1.1, while 1,250 examples are synthesized to enhance LLaMA-3 due to denser knowledge in it.". Why do you use different numbers for LLaMA-3?
7. "Therefore, Single enhances Mistral, Qwen2, and Gemma-1.1 with 1,500 examples, and it utilizes 600 examples to enhance LLaMA-3.". Is it fair to use different number of examples for LLaMA?
8. "Naive and Single could supplement some knowledge to them but cause them to forget more useful knowledge." Your method also supplies some knowledge but it does not hurt the numbers. why single hurts? how do you select the example for single one?
9. In section 3.6, item (2), you claim that “Naive and Single could supplement some knowledge to them but cause them to forget more useful knowledge.”. why this forgetting does not happen in LaMer?

---

> ### Author Response · Authors · 2025-11-21
>
> We would like to sincerely thank you for the careful reading of our manuscript and for the constructive comments.
> ## Weaknesses: This method is limited to adding knowledge from GenericsKB. They do not show how much knowledge in GenericsKB is actually missing in the LLM they study (maybe for some of them, the knowledge is already there, but fine-tuning only makes that knowledge sharp). Heavy reliance on ChatGPT (if ChatGPT makes errors, their method will as well).
> We focus on the knowledge deficiency in LLMs, which contain several types of them (lines 166-171): (1) Helpful: LLMs do not know this knowledge, or they cannot use the knowledge sharply. (2) Misleading: LLMs are easily misled by the knowledge (maybe irrelevant). Furthermore, knowing how much knowledge LLMs do not know is the task of knowledge probing, while we focus on “deficiency”.
>
> Due to resource and time constraints, we have explored using Qwen3-32B to synthesize data for comparison. Results on Mistral are shown in the following table. We can find that changing gpt-3.5 to Qwen3-32B can result in a small performance gain, which demonstrates the robustness of our method with open-source models.
> | Method                          | Comm. | AGIEval |  ARC  | MMLU  |  BBH  | CRASS | GSM-Plus | Average |
> | :------------------------------ | :---: | :-----: | :---: | :---: | :---: | :---: | :------: | :-----: |
> | LaMer (Mistral)  with GPT-3.5   | 75.10 |  34.50  | 77.22 | 54.52 | 33.72 | 88.33 |  29.23   |  56.09  |
> | LaMer (Mistral)  with Qwen3-32B | 74.97 |  35.20  | 77.20 | 53.69 | 34.16 | 86.67 |  30.95   |  **56.12**  |
>
> Additionally, using a strong LLM to guide smaller ones is what many researchers do, such as Qwen3 and DeepSeek, and we do agree that ChatGPT makes errors. Although ChatGPT makes errors, our method can still bring improvement to LLMs, which further demonstrates the robustness of our method. If ChatGPT improves, our method can also improve.
> ## Question 1
> The knowledge will be provided to ChatGPT, which means we do not require ChatGPT to “know” the knowledge. Furthermore, the query will be provided to ChatGPT to help understand the knowledge.
>
> Additionally, ChatGPT is much stronger and has much broader knowledge than our investigated LLMs, and we follow previous research to use strong LLMs (ChatGPT) to enhance smaller ones [1][2].
>
> [1] Meaningful Learning: Enhancing Abstract Reasoning in Large Language Models via Generic Fact Guidance
>
> [2] Mixed Distillation Helps Smaller Language Models Reason Better
> ## Question 2
> We suppose the reasons are twofold:
> (1) Gemma-1.1 poses high knowledge density, which is defined as pretraining tokens/parameters. Gemma-1.1-2B has a T/P ratio of over1500:1 (trained on 3T tokens) [1], while Qwen2-7B has T/P ratio lower than 1000:1 (trained on 7T tokens) [2]. Therefore, baselines can cause the loss of more knowledge while injecting new knowledge, resulting in worse performance in baselines. However, our target improvement can acquire more knowledge than losing, which might lead to more extensive improvement in Gemma-1.1 than in others.
>
> (2) Gemma-1.1 is the smallest one among all investigated LLMs, which makes our targeted improvement more extensive.
>
> [1] Gemma: Open Models Based on Gemini Research and Technology
>
> [2] QWEN2 TECHNICAL REPORT
> ## Question 3
> In this context, “valid” refers to examples that have an answer, as some examples in GSM-Plus lack ground-truth answers (answer is None in the example).
> ## Question 4
> Please refer to the response to the weakness.
> ## Question 5
> Sorry for the mistake, the size of each group on LLaMA-3 is 125, we have corrected it in the revised version.
> ## Question 6
> LLaMA-3 has much denser knowledge than other LLMs (even denser than Gemma-1.1), with a token/parameter ratio of 1875:1, which could result in more disturbance when introducing more new knowledge. Thus, we only use 1,250 examples to enhance LLaMA-3.
> ## Question 7
> We focus on performance across methods within the same LLM and do not need to compare performance across LLMs. The reasons are: (1) we use different LLMs for experiments to demonstrate the general applicability of our method. (2) Methods using different base LLMs are not directly comparable, as the underlying base LLMs heavily influence performance.
> ## Question 8
> Single keeps only one example for each deficiency in our method. Single do not perform better than baselines because only one example is insufficient to remedy the severe deficiencies, this also explains why we should synthesize more examples for severer deficiencies.
> ## Question 9
> LaMer focuses on target improvement, which means LaMer could inject much more useful knowledge than baselines and will not cause overfitting on memorized knowledge. Additionally, baselines do not contain enough knowledge or the examples are insufficient to remedy deficiencies well, which causes they to lose more knowledge than they obtain.

---

### Official Review · Reviewer_jb7t · 2025-11-01

**Soundness:** 3
**Presentation:** 3
**Contribution:** 3
**Rating:** 4
**Confidence:** 3

**Summary:**

This paper introduces a label-free framework for identifying and improving knowledge gaps in large language models (LLMs), which does not rely on costly human annotations. Specifically, the authors propose the relative-entropy-based diagnostic method that quantifies how much additional information external knowledge contributes to the LLM, which leads to detecting the areas of weakness. Then, the authors design the remedy process based on the curricular learning: synthesizing examples in proportion to deficiency severity and training the model from easier to harder deficiencies. The authors validate the proposed approach on four LLMs and multiple reasoning benchmarks, showing that it achieves consistent performance gains across them.

**Strengths:**

* The processes to identify and remedy deficiencies in LLMs are convincing.
* The proposed approach clearly outperforms existing relevant baselines.

**Weaknesses:**

* In extracting the knowledge (needed to check and remedy deficiencies in LLMs), the assumption that, for each query, there should be relevant knowledge from an external knowledge base is very strong. In other words, what if the external knowledge base does not contain the relevant knowledge for each query? Additionally, the process of checking and remedying knowledge deficiencies can be done only for knowledge within the knowledge base, which seems a clear limitation of the proposed approach. Lastly, I am a bit confused whether the proposed approach is truly label-free: it requires the knowledge that is related and associated with the query, which may be considered as the label for the query.
* There are relevant papers [A, B, C] that the authors should discuss and potentially compare with, especially [A] (which seems highly relevant).
* The authors could more explicitly justify the advantage of the proposed approach over the unsupervised learning (or SFT) with experiments (i.e., the current setup does not fully justify its advantage over them, despite the claims in the paper). For example, the authors could train the LLMs with all the knowledge in the whole knowledge base and compare the proposed approach against it (i.e., the unsupervised setup) in both effectiveness and efficiency.
* The performance of the baseline approaches on the Gemma-1.1 (2B) is inferior to the most basic setup (called Base), which may warrant more discussions.

---

[A] Structural Entropy Guided Agent for Detecting and Repairing Knowledge Deficiencies in LLMs, 2025.

[B] R-Zero: Self-Evolving Reasoning LLM from Zero Data, 2025.

[C] Self-Error-Instruct: Generalizing from Errors for LLMs Mathematical Reasoning, 2025.

**Questions:**

Please see Weaknesses above.

---

> ### Author Response · Authors · 2025-11-21
>
> We would like to sincerely thank you for the careful reading of our manuscript and for the constructive comments.
>
> ## Weakness 1-1: What if the external knowledge base does not contain the relevant knowledge for each query? Additionally, the process of checking and remedying knowledge deficiencies can be done only for knowledge within the knowledge base, which seems a clear limitation of the proposed approach.
>
> We have conducted an extra experiments to demonstrate the robustness of our method when facing retrieval noise.
>
> We have replaced the knowledge retrieval process with a random sampling process. For each query, we perform uniform sampling over the knowledge base to obtain four pieces of knowledge. After that, we follow the deficiency diagnosis step to compute RE. The following table shows the distribution of the RE in our method and the random sampling method.
> |Method| Easy  | Normal | Hard | Unfair |
> |-|:---:|:---:|:--:|:----:|
> |**Random**|3,182|937|322|556|
> |**Ours**|10,973|3,683|1,971|5,926|
> |**[Random/Ours] Ratio**|0.29|0.25|0.16|0.09|
>
> We can find: **(1) the retrieval noise does not impact too much on RE computation. (2) The higher the RE , the smaller the impact of noise on RE.** This demonstrates that our method is robust to retrieval noise.
>
> Additionally, our method supports all kinds of knowledge bases, such as web text and wiki. Therefore, we can always find related knowledge for the given query.
>
> ## Weakness 1-2: Whether the proposed approach is truly label-free, since it requires the knowledge that is related and associated with the query, which may be considered as the label for the query.
> We suppose “knowledge” is far from being considered as labels. First, labels are limited and require annotation, while knowledge is easy to obtain and is everywhere. Second, labels can indicate the correctness of LLMs, while knowledge is merely a contextual intervention that influences the model’s internal reasoning.
>
> ## Weaknesses 2: There are relevant papers [A, B, C] that the authors should discuss and potentially compare with, especially [A] (which seems highly relevant).
> Thanks for your advice. We are willing to discuss these papers in the final version due to the page limit. For example, there are three key differences between [A] and our method:
>
> (1) Label-free diagnosis: While [A] leverages structural entropy and knowledge graphs (KGs) to navigate and identify deficiencies, our method (LaMer) uses relative entropy on model distributions in a fully label-free setting, thereby removing the need for labeled correctness or domain-specific supervision. (2) Knowledge source & generality：[A] assumes and relies on a structural domain-specific KG and uses MCTS to explore that structured knowledge space. In contrast, LaMer works with more general knowledge retrieval (e.g., from GenericsKB) and is designed for broader, less constrained domains and queries. (3) Remedy strategy & curriculum design：[A] focuses on targeted data generation via graph exploration, while LaMer introduces a curricular meaningful learning framework, which enables efficient remedy even with limited synthetic data (e.g., only 40% training data used).
>
> These differences mean that our method cannot be directly compared with [A], and we will clarify these distinctions explicitly in the final version due to the page limit.
>
> ## Weaknesses 3: The authors could more explicitly justify the advantage of the proposed approach over the unsupervised learning with experiments (despite the claims in the paper). For example, the authors could train the LLMs with all the knowledge in the whole knowledge base and compare the proposed approach against it in both effectiveness and efficiency.
> We follow the conventional methods to choose baselines for comparisons in more realistic settings. On the one hand, the whole knowledge base contains over 3.4M pieces of knowledge, training LLMs with the full knowledge base is costly and time-consuming. On the other hand, “Naive” method uses the same size of training data as our proposed LaMer. LaMer can outperform Naive on four LLMs across 7 OOD benchmarks, we suppose this sufficiently demonstrates the effectiveness of our method. Single can achieve similar performance with Naive, which further demonstrates the efficiency.
> ## Weakness 4: The performance of the baseline approaches on the Gemma-1.1 (2B) is inferior to the most basic setup (called Base), which may warrant more discussions.
> We suppose this comes from the high destiny of knowledge in Gemma-1.1, which is defined as pretraining tokens/parameters. For example, Gemma-1.1-2B has a T/P ratio of over 1500:1 (trained on 3T tokens) [1], while Qwen2-7B has a T/P ratio lower than 1000:1 (trained on 7T tokens) [2]. Therefore, baselines can cause the loss of more knowledge while injecting new knowledge. We will discuss this in the final version due to the page limit.
>
> [1] Gemma: Open Models Based on Gemini Research and Technology
>
> [2] QWEN2 TECHNICAL REPORT

---

### Author Response · Authors · 2025-12-02
**Rebuttal Summary & Final Remarks**

Dear Reviewers and ACs,

Thanks for your time and effort in reviewing our paper. Here, we briefly summarize the rebuttal status:

We are glad that the reviewers recognize the strengths and contributions of our work. Specifically highlighted by reviewers jb7T and 4XWx, our method is **convincing**, **well-motivated**, and **novel**. Reviewers 85W6 and oqoY appreciated that our approach **eliminates the need for human annotation or labels**, which both **simplifies practical adoption and broadens applicability**. All reviewers also emphasized both the **effectiveness** and **efficiency** of our approach. Moreover, reviewers 85W6 and oqoY found the paper clearly written and easy to follow. Our method innovatively used relative entropy to diagnose LLMs' deficiencies under a label-free setting and remedy them with curricular meaningful learning, thereby ensuring the coverage and eliminating the need for human annotation. Overall, our method outperforms baselines in OOD benchmarks in both effectiveness and efficiency, which is supported by results across different model families, providing finer-grained insights.



We believe we have addressed the main concerns of the reviewers:

- **Regarding Robust to Retrieval Noise** [reviewers jb7T, 4XWw, and oqoY]: the reviewers are concerned that the retrieval noise would have a significant impact on the RE computation (deficiency diagnosis) in our method. To address this concern, we have experimented by replacing the knowledge retrieval process with a random sampling process. From the deficiency diagnosis results below, we can find: **(1) the retrieval noise does not impact RE computation. (2) The higher the RE , the smaller the impact of noise on RE.** This demonstrates that our method is robust to retrieval noise.
| Method                  |  Easy  | Normal | Hard  | Unfair |
| ----------------------- | :----: | :----: | :---: | :----: |
| Random              | 3,182  |  937   |  322  |  556   |
| Ours                | 10,973 | 3,683  | 1,971 | 5,926  |
| [Random/Ours] Ratio |  0.29  |  0.25  | 0.16  |  0.09  |

- **Regarding Sensitivity of RE Thresholds** [reviewers 4XWw and oqoY]: the reviewers are concerned that the performance of our method might be sensitive to the RE thresholds, thus influencing the application of our method. To address this concern, we have tried two different kinds of RE thresholds (Value-based and Quantile-Based) to investigate the sensitivity of RE thresholds. From the result bwlow, our method can achieve stable performance with different RE thresholds, which demonstrates the robustness and stabilization of our proposed method on different RE threshold choices.
| Method                          | Comm. | AGIEval | MMLU  | Average |
| ------------------------------- | :---: | :-----: | :---: | :-----: |
| Mistral (Base)                  | 67.56 |  32.82  | 49.75 |  50.04  |
| LaMer with Value-based Grouping | 75.26 |  35.24  | 54.14 |  54.88  |
| LaMer Quantile-Based Grouping   | 74.94 |  34.93  | 53.73 |  54.53  |
| LaMer (Ours)                    | 75.10 |  34.50  | 54.52 |  54.71  |

- **Regarding Dependence on a Closed-source LLM—ChatGPT** [reviewers 85W6 and 4XWw]: reviewers are concerned that the reliance on closed-source LLM (ChatGPT) would lead to limited applicability. To address this concern, we replace ChatGPT with an open-source LLM Qwen3-32B for data synthesis. Results below show that Qwen3-32B could achieve similar or even slightly better performance than ChatGPT, which demonstrates the robustness of our method with open-source models.
| Method                          | Comm. | AGIEval |  ARC  | MMLU  |  BBH  | CRASS | GSM-Plus |  Average  |
| :------------------------------ | :---: | :-----: | :---: | :---: | :---: | :---: | :------: | :-------: |
| LaMer (Mistral)  with GPT-3.5   | 75.10 |  34.50  | 77.22 | 54.52 | 33.72 | 88.33 |  29.23   |   56.09   |
| LaMer (Mistral)  with Qwen3-32B | 74.97 |  35.20  | 77.20 | 53.69 | 34.16 | 86.67 |  30.95   | **56.12** |

- **Clarification, Explanation, and Discussion**: we have provided detailed clarification and explanation for the differences between knowledge and labels [reviewer jb7T], the differences between our paper and related works [reviewer jb7T], the performance of Gemma-1.1 and Qwen2 [reviewer jb7T and 856W], as well as the adaption of our method in private, streaming, or domain-restricted settings [reviewer oqoY].



We have also revised the equation in the revised manuscript based on suggestions from reviewer 4XWw.

Regrettably, there was no further engagement from the reviewers during the discussion period, which is particularly unfortunate given that reviewer jb7T appear to have misunderstandings regarding the paper. We hope the AC will consider our detailed responses.

Best regards,

The Authors

---

### Meta-Review · Area_Chair_YeTK · 2026-01-07

**Summary:**

This paper introduces LaMer, a well-motivated and label-free framework for diagnosing and remedying knowledge deficiencies in LLMs using relative-entropy–based analysis and curriculum-driven data synthesis. Reviewers found the approach sound, effective, and practically valuable, with consistent gains across multiple models and benchmarks. To further improve the work, the authors could add discussions on robustness to knowledge base coverage and retrieval noise, and provide additional comparisons with more baselines.

**Reviewer Concerns:**

Reviewers are generally positive about this work and would like to see more discussions on robustness to knowledge base coverage and retrieval noise, and comparisons to more baselines.

**Reviewer Scores:**

Reviewer generally keep their scores after rebuttal.

---

### Decision · Program_Chairs · 2026-01-26

Accept (Poster)